# Curcumin modulated gut microbiota and alleviated renal fibrosis in 5/6 nephrectomy-induced chronic kidney disease rats

**Cheng Li**[1,2,3], **Xulong Chen**[4], **Jingchun Yao**[3], **Weiwei Zha**[4], **Meiren Li**[5], **Jiangwen Shen**[4], **Hongli Jiang**[6], **PuXun Tian**[1,2]*

1 Department of Kidney Transplantation, Nephropathy Hospital, The First Affiliated Hospital of Xi'an Jiaotong University, Xi'an, Shaan'xi, China, 2 Institute of Organ Transplantation, Xi'an Jiaotong University, Xi'an, Shaan'xi, China, 3 Department of Nephrology, Jiujiang University Affiliated Hospital, Jiu'jiang, Jiang'xi, China, 4 School of Clinical Medical, Jiujiang University, Jiu'jiang, Jiang'xi, China, 5 Department of Pathology, Jiujiang University Affiliated Hospital, Jiu'jiang, Jiang'xi, China, 6 Department of Nephrology, Xi'an Jiaotong University, Xi'an, Shaan'xi, China

* yuantian@mail.xjtu.edu.cn

**Data Availability Statement:** All relevant data are within the manuscript and its Supporting Information files. All gut microbiome dataset files

## Abstract

Increasing evidence suggests that dysbiosis of gut microbiota exacerbates chronic kidney disease (CKD) progression. Curcumin (CUR) has been reported to alleviate renal fibrosis in animal models of CKD. However, the relationship between CUR and gut microbiome in CKD remains unclear. This study aims to investigate the potential anti-renal fibrosis effects of CUR from the gut microbiota perspective. A 5/6 nephrectomy (5/6Nx) rat model was used to explore the therapeutic effect of CUR on renal fibrosis. Tight junction protein expression levels were measured to assess intestinal barrier function. 16S rRNA sequencing was employed to evaluate changes in gut microbiota composition, and metabolomics was utilized to detect alterations in plasma metabolites. The administration of CUR significantly ameliorated renal fibrosis and inhibited inflammation in 5/6Nx rats. Additionally, CUR markedly improved the expression of tight junction proteins and local colon inflammation. CUR also positively reconstructed gut microbiota, significantly increasing the abundance of beneficial bacteria, such as *Lachnospiraceae_NK4A136_group*, *Eubacterium_siraeum_group*, and *Muribaculaceae* was significantly increased. Metabolomics revealed that CUR reduced uremic retention solutes and elevated Vitamin D and short-chain fatty acids (SCFAs). Spearman correlation analysis indicated that gut genera enriched by CUR were positively correlated with Vitamin D and SCFA and negatively correlated with chronic renal injury biomarkers. Mechanistically, we found inhibition of the LPS/TLR4/NF-κB and TGF-β1/Smads pathway in CUR-treated rats. Our study indicates that CUR has the potential to modulate gut microbiota composition, and that this modulation may contribute to the anti-fibrosis effects of CUR.

are available from the NCBI database (accession number(s): PRJNA1069977).

**Funding:** This project was assisted by National Natural Science Foundation of the People's Republic of China (82270791), Jiangxi Traditional Chinese Medicine Science and Technology Plan Project (2022A137, 2023B1287), Jiangxi Provincial Health Commission Science and Technology Program (202212007), Science and Technology Research Project of Jiangxi Provincial Department of Education (GJJ201819, GJJ211805), Beijing Medical and Health Foundation (TYU046B) and Beijing Medical Award Foundation (YXJL-2022-0734-0294).

**Competing interests:** The authors have declared that no competing interests exist.

# Introduction

Chronic kidney disease (CKD) prevalence has steadily increased over the past few decades due to a number of factors. The Lancet journal published a research report on global CKD from 1990 to 2017. According to the report, there were 697.5 million patients with CKD worldwide in 2017, with China having the highest number of patients at 132.3 million [1]. Renal fibrosis contributes significantly to the development of CKD, which in turn leads to end-stage kidney disease (ESKD). When stressed or injured, the intrinsic cells of the kidney (tubular epithelial cells, endothelial cells, fibroblasts) and macrophages infiltrating the kidney are involved in the progress of renal fibrosis [2]. Myofibroblasts proliferate and produce large quantities of extracellular matrix (ECM). Excessive ECM contributes to abnormal fibrosis, resulting in renal impairments [3]. Thus, early intervention and control of renal fibrosis are crucial in the treatment of CKD. Currently, the main CKD management approaches include nutritional intervention, lifestyle change, and blood pressure control [4, 5]. Aside from kidney transplantation, there seems to be no available effective treatment to halt the process of renal fibrosis. Consequently, the search for better treatment strategies for CKD has become a current research hotspot.

Gut microbiota is a complex community in the human gut, which profoundly impacts the physiological process or pathological changes of the human body, while dysregulation of the gut microbiota can exacerbate the progression of various diseases [6]. Increasing evidence suggests that gut microbiota plays a key role in the pathogenesis and development of CKD [7]. During the progression of CKD, gut microbiome dysbiosis results in the generation of microbial-derived urinary toxins. This disruption of the intestinal epithelial barrier function allows gut-derived urinary toxins and bacterial DNA to enter the systemic circulation, further aggravating the progression of CKD [8]. Lipopolysaccharides (LPSs) are produced by Gram-negative bacteria, which derive from altered gut microbiota and intestinal leakage and are amongst the most pro-inflammatory molecules known [9]. When the intestinal mucosal barrier function is disrupted, LPS crosses the intestinal barrier and enters the blood circulation to the kidney. It binds to the Toll-like receptors (TLRs) of the innate immune system and becomes an important mediator of kidney injury [10]. NF-κB is a ubiquitous transcription factor. After activation by TLR4, NF-κB plays an important role in controlling the expression of pro-inflammatory genes and immune system-related genes, promoting inflammatory factors expression, cell adhesion molecular proteins, and chemokines, thus accelerating the formation of fibrosis [11, 12]. Hence, targeting the gut microbiota may be a new approach to treating renal fibrosis in CKD.

Curcumin (CUR), the principal active constituent of turmeric, exhibits a broad spectrum of pharmacological activities and has garnered significant attention as a widely utilized natural product. CUR has been proven to have anti-inflammatory [13], antibacterial [14], anti-fibrosis [15], and immune regulatory effects [16] and has been reported to be widely used in various chronic diseases, such as autoimmune diseases, tumors, cardiovascular and neurological diseases, etc. In recent years, the regulatory effects of CUR on intestinal microecology have been extensively studied [17]. It has been reported that such polyphenols preferentially distribute and accumulate in the intestinal tract and can directly act on the gut microbiota to exert their pharmacological effects [17]. Currently, research on the treatment of CUR in CKD is advancing. Many animal and cellular experiments have used CUR to prevent and treat various kidney diseases, and even some preliminary reports on CUR reshaping the gut microbiota of CKD [18, 19]. According to a clinical study, after supplementing CUR, the microbial diversity of CKD patients showed a significant trend similar to that of healthy individuals. After 6 months of therapy, *Escherichia-Shigella* was significantly lower, while *Lachnoclostridium* was

significantly higher [19]. Another experimental study of CUR on rats with uric acid nephropathy (UAN) showed that CUR treatment protected against the overgrowth of opportunistic pathogens in UAN, including *Escherichia-Shigella* and *Bacteroides*, and increased the relative abundance of bacteria producing short-chain fatty acids (SCFAs), such as *Lactobacillus* and *Ruminococcaceae* [20]. However, there is currently a lack of relevant research investigating the effect of CUR on gut microbiota and bacteria-derived metabolites in the renal fibrosis model.

In this study, we reported that CUR ameliorated kidney function and renal fibrosis in 5/6 Nx rats. The underlying mechanism may be the LPS/TLR4/NF-κB signaling pathway inhibited by CUR. Moreover, CUR improves intestinal integrity and enhances the intestinal mucosal barrier. Research on gut microbiota and metabolomics analysis showed that CUR regulated gut microbiota disorder and elevated the contents of Vitamin D and SCFA. Therefore, our findings further demonstrate that CUR acts as a prebiotic in the treatment of renal fibrosis (Fig 1).

## Materials and methods

### Ethics approval

The animal study was reviewed and approved by the Animal Care and the Medical Ethics Committee of Jiujiang University Affiliated Hospital, Jiujiang, China (protocol approval number: jjumer-a-2021-0701; approval date: 20 April 2021).

### Reagents

Curcumin (purity 96.18%, batch number JHS20180328) was provided by Huisheng Biotechnology Co., Ltd (Xi'an, China).

### Animals

Specific pathogen-free (SPF) grade male SD rats (6–8 weeks old, 180-220g) were purchased from the experimental centre of Jiangxi University of Traditional Chinese Medicine (Nanchang, China) [SCXK (Gan) 2018–0003]. SD rats were raised at the animal experiment center of Jiujiang University [SYXK (Gan) 2017–001]. Approximately one week prior to the experiment, the rats were transferred to a laboratory environment for acclimatization to the feeding regimen. During this acclimation period, the animals had ad libitum access to both water and food. The diet administered to the rats in each group comprised a standard feed formulation consisting of 73.5% corn, 20% wheat bran, 5% fish meal, 1% cereal flour, and 0.5% salt. The standard experimental environment was 25 ± 2˚C, 50–60% humidity, and alternating 12h light-dark periods. Experiments were performed following the 3R principle (reduction, replacement, and refinement) to avoid or minimize pain, and the animals were handled calmly by trained personnel.

### Model establishment and experimental design

After one week of adaptive feeding, SD rats were randomly divided into sham (n = 9)and model groups(n = 18). The model group underwent 5/6 nephrectomy (5/6Nx) according to the previously reported method [21, 22] to create a CKD rat model. The sham group underwent two-stage open surgeries without Nx. The specific surgical process was as follows:

Anesthesia was induced with inhaled 4% isoflurane before surgery and maintained with 2.5% isoflurane during animal surgery. The backs of rats were depilated and prepared for surgery. Anesthetized rats were fixed in a prone position on a mouse board, the skin preparation area was disinfected with 75% alcohol, and sterile surgical drapes (self-made gauze) were

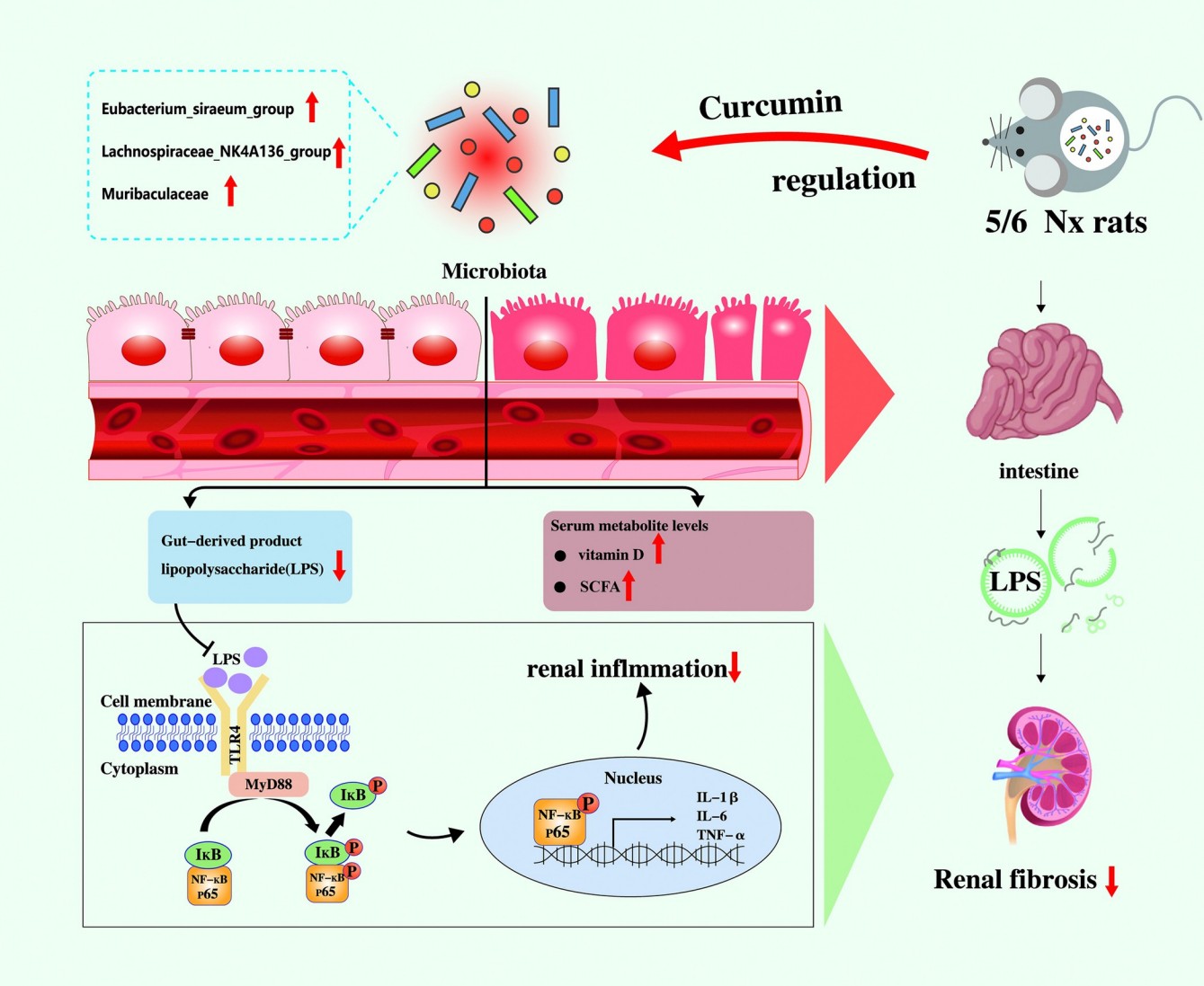

**Fig 1. Schematic flowchart of the study design.**

placed. A longitudinal 2 cm surgical incision was made next to the spine at the lower left costal margin; the skin and subcutaneous tissue were cut layer by layer, and the muscle was bluntly separated. Next, the perirenal fat sac was gently lifted with forceps, placed the surgical handle on it, exposed and fixed the kidney outside the incision, peeled off the renal capsule and perirenal fat, exposed the upper and lower poles of the kidney, used a sharp blade to remove the upper and lower poles of the kidney, and quickly compressed the renal section with gelatin sponge for 30 seconds to stop bleeding, then returned the kidney and sutured layer by layer. One week later, the right kidney was exposed using the same method. After peeling off the renal capsule and perirenal fat, the 4–0 silk thread was used to ligate the renal pedicle along the renal hilum. Then, the right kidney was removed and sutured layer by layer after confirming no bleeding. The sham procedure was performed by only peeling off the renal membrane without damaging the kidney. The other steps were the same as above. After the operation, the rats were fed and watered freely in an SPF environment at room temperature and 45% relative humidity.

After the 5/6 nephrectomy, rats in the model group were randomly assigned to the 5/6 Nx group and the 5/6Nx+CUR group (n = 9 per group). The 5/6Nx+CUR group was given CUR suspension (200 mg/kg, in 0.1% CMC-Na) by gavage once per day for 12 weeks. The sham and 5/6Nx groups received the same amount of 0.1% CMC Na. The dose of CUR was based on the human intake level according to the equivalent dosage conversion coefficient between rats and humans.

One day before sacrifice, all rats were placed in metabolic cages and fasted for 24 hours with free access to water. We collected 24-hour urine volume and fresh faeces the next day, followed by the use of isoflurane as an inhalant anesthetic to euthanize the rats. After that, we collected blood samples from the posterior orbital venous plexus. The above urine, faeces, and serum samples were stored in a -80˚C refrigerator for testing. The kidney, ileum, and colon tissues were separated into 2 parts: the first was fixed with 4% formaldehyde for pathological tissue sectioning, and the second was stored in a -80˚C refrigerator for subsequent study.

## Histopathology analysis in kidney and intestinal tissues

The kidney and intestinal tissues fixed with formalin solution, dehydrated and embedded to make pathological sections. Sections of kidney tissues (4 μm thick) were stained with Masson's trichrome, Sirius red, and periodic acid-Schiff (PAS) staining. Masson image analysis and collagen volume fraction (CVF, %) were determined using Image Pro Plus 6.0. Tubular injury scores were evaluated in terms of tubular dilatation, tubular necrosis, cell necrosis, and brush edge loss. Specific scoring criteria: 0 points, normal; 1 score, ≤25%; 2 points, 25% ~ 50%; 3 points, 50% ~ 75%; 4 points, 75% to 100%. Intestinal tissues (4 μm thick) were investigated after hematoxylin-eosin (H&E) and PAS staining. The histopathological score of the ileum and colon was performed according to pathological scoring criteria. Ten randomly selected non-overlapping areas were acquired with light microscopy (Olympus BX51, Olympus, Japan).

## Biochemical testing and enzyme-linked immunosorbent assay (ELISA) detection

Serum creatinine (Scr) and urea nitrogen (BUN) levels were determined using a BECKMAN coulter AU5831 automatic biochemical analyzer. Urinary protein was detected using the BCA (Bicinchonic acid) protein concentration assay kit (Beckman Coulter, Brea, CA, USA). The 24h urine protein quantitation (24H-P, g/24h) was calculated as urine protein content (g/L) × total 24-hour urine volume (L). Monocyte chemoattractant protein-1 (MCP-1) and LPS concentrations were assessed using an ELISA kit (Meimian, Jiangsu, China) in accordance with the guidelines provided by the manufacturer.

## RNA extraction and real-time PCR

Total RNA was isolated from kidney and ileum tissues by referring to trizol reagent instructions (TransGen Biotech, Beijing, China). RNA concentration and purity were measured using the Novogene spectrophotometer (US850, Beijing, China). Reverse transcription of RNA into cDNA was performed using a 20μL reaction system (TransGen Biotech, Beijing, China). mRNA gene expression was measured by SYBR Green quantitative real-time PCR on a cDNA template (CWBIO, Beijing, China). The following conditions were used for the reaction: 42˚C for 1 h, 25˚C for 5 min, and 70˚C for 5 min. Then, cDNA was used as the template for RT-PCR amplification. The following were the reaction conditions: predenaturation at 95˚C for 3 min, denaturation at 95˚C for 10 s, and annealing at 58˚C for 30 s, and extension at 68˚C for 20 s with 40 amplification cycles. Experiments were repeated three times for each sample. GAPDH was used as an internal control gene, and the relative expression level of the

target gene was calculated using the $2^{-\Delta\Delta CT}$ method. The primers used in this study include *FN*, *ACTA2*, *COLIA1*, *IL-1β*, *IL-6*, *TNF-α*, *MCP1*, *Ocln*, *Cldn1*, *ZO-1* and primer sequences are given in S1 Table.

## Western blot analysis

Total protein was extracted from the rat's kidney, ileal, and colon tissue lysate using RIPA buffer (Solarbio, Beijing, China) and 1%PMSF (Beyotime, Shanghai, China). Protein concentration was determined using a BCA kit (PICPI23223, Thermo Fisher, USA). And total protein samples were separated by SDS-PAGE gel electrophoresis and transferred to PVDF membranes for subsequent analysis (Millipore, USA). The membrane was blocked with 5% skim milk in Tris-buffered saline with 1% Tween 20 (TBS-T) for 1 h. Then, the membrane was incubated with primary antibodies at 4°C overnight, followed by incubation with the corresponding secondary antibodies. β-actin was used as an internal control. The expression level of the protein was detected by chemiluminescence (Tanon-4800, Shanghai, China), and the band intensity was quantified using ImageJ software. The antibodies used for blotting are as follows: Transforming growth factor beta (TGF-β1; HA721143, 1:500 dilution), Smad 2 (ET1604-22, 1:1000 dilution) and Smad 3 (ET1607-41, 1:1000 dilution) were provided by HuaBio (Hangzhou, China); TLR4 (66350-1-Ig, 1:1000 dilution), zonula occludens 1 (ZO-1; 21773-1-AP, 1:5000 dilution), occludin (27260-1-AP, 1:5000 dilution), claudin-1 (28674-1-AP, 1:1000 dilution) and IKBα (10268-1-AP, 1:5000 dilution) were purchased from Proteintech (Wuhan, China) and Phosphorylated (p)-P65 (AF2006, 1:1000 dilution) was acquired from Affinity Biosciences (Cincinnati, USA). The antibody against P65 (8242S, 1:1000 dilution) was procured from Cell Signaling Technology (MA, USA). p-IKBα (bsm-52169R, 1:1000 dilution) was from BIOSS (Beijing, China) and anti-β-actin antibody (AC026, 1:50000 dilution) was procured from ABclonal Technology (MA, USA).

## 16S rDNA gene sequencing

Total genomic DNA was extracted using MagPure Soil DNA LQ Kit (Magan) following the manufacturer's instructions. DNA concentration and integrity were measured with NanoDrop 2000 (Thermo Fisher Scientific, USA) and agarose gel electrophoresis. Extracted DNA was stored at -20°C until further processing. The extracted DNA was used as a template for PCR amplification of bacterial 16S rRNA genes with the barcoded primers and Takara Ex Taq (Takara). For bacterial diversity analysis, V3-V4 (or V4-V5) variable regions of 16S rRNA genes were amplified with universal primers 343F (5′–TACGGRAGGCAGCAG–3′) and 798R (5′–AGGGTATCTAATCCT–3′) for V3-V4 regions. The Amplicon quality was visualized using agarose gel electrophoresis. The PCR products were purified with AMPure XP beads (Agencourt) and amplified for another round of PCR. After being purified with the AMPure XP beads again, the final amplicon was quantified using the Qubit dsDNA Assay Kit (Thermo Fisher Scientific, USA). The concentrations were then adjusted for sequencing. The sequencing was performed on an Illumina NovaSeq 6000 with 250 bp paired-end reads (Illumina Inc., San Diego, CA; OE Biotech Company, Shanghai, China). Raw sequencing data were in FASTQ format. Paired-end reads were then preprocessed using cutadapt software to detect and cut off the adapter. After trimming, paired-end reads were filtered for low-quality sequences, denoised, merged, and detected and cut off the chimera reads using DADA2 with the default parameters of QIIME2 (2020.11). Finally, the representative reads and the ASV abundance table was generated by the software. The representative read of each ASV was selected using the QIIME2 package. All representative reads were annotated and blasted

against Silva database Version 138 (or Unite) (16s/18s/ITS rDNA) using q2-feature-classifier with the default parameters (S1 File).

## LC/MS untargeted metabolomics analysis

80 μL of sample was added to a 1.5 mL Eppendorf tube with 10 μL of L-2-chlorophenylalanine (0.3 mg/mL) dissolved in methanol as internal standard, and the tube was vortexed for 10 s. Subsequently, 240 μL of an ice-cold mixture of methanol and acetonitrile (2/1, vol/vol) was added, the mixtures were vortexed for 1 min, and the whole samples were extracted by ultrasonication for 10 min in an ice-water bath, and stored at -20˚C for 30 min. The extract was centrifuged at 4˚C (13,000 rpm) for 10 min. 160 μL of supernatant in a glass vial was dried in a freeze concentration centrifugal dryer. Then, 240 μL mixture of methanol and water (1/4, vol/vol) were added to each sample, the samples vortexed for 30 s, extracted by ultrasonication for 3 min in ice-water bat, and then placed at -20˚C for 2 h. The samples were then centrifuged at 4˚C (13,000 rpm) for 10 min. The supernatants (150 μL) from each tube were collected using crystal syringes, filtered through 0.22 μm microfilters, and transferred to LC vials. Finally, 15 μL supernatant was used for LC-MS detection.

The original LC-MS data were processed by the software Progenesis QI V2.3 (Nonlinear, Dynamics, Newcastle, UK) for baseline filtering, peak identification, integral, retention time correction, peak alignment, and normalization. The main parameters of 5 ppm precursor tolerance, 10 ppm product tolerance, and 5% product ion threshold were applied. Compound identification was based on the precise mass-to-charge ratio (M/z), secondary fragments, and isotopic distribution using The Human Metabolome Database (HMDB), Lipidmaps (V2.3), Metlin, EMDB, PMDB, and self-built databases to do qualitative analysis. The extracted data were then further processed by removing any peaks with a missing value (ion intensity = 0) in more than 50% of groups, replacing the zero value with half of the minimum value, and screening according to the qualitative results of the compound. Compounds with scores below 36 (out of 60) points were also deemed inaccurate and removed. A data matrix was combined from the positive and negative ion data. The matrix was imported in R to carry out Principal Component Analysis (PCA) to observe the overall distribution among the samples and the stability of the whole analysis process. Orthogonal Partial Least-Squares-Discriminant Analysis (OPLS-DA) and Partial Least-Squares-Discriminant Analysis (PLS-DA) were utilized to distinguish the metabolites that differ between groups. To prevent overfitting, 7-fold cross-validation and 200 Response Permutation Testing (RPT) were used to evaluate the quality of the model. Variable Importance of Projection (VIP) values obtained from the OPLS-DA model were used to rank the overall contribution of each variable to group discrimination. A two-tailed Student's T-test was further used to verify whether the metabolites of difference between groups were significant. Differential metabolites with VIP values greater than 1.0 and $p$-values less than 0.05 were selected (S1 File).

## Statistical analysis

All the data (except for data analysis of gut microbiota and metabolomics) were analyzed with GraphPad Prism 8 (GraphPad Software, USA) statistical analysis software and expressed as the mean ± SD. The measurement data were assessed for adherence to normal distribution. The data normality test was conducted using the Shapiro-Wilk test. The mean value of each group was analyzed using a one-way ANOVA analysis with Tukey post-hoc tests. $P < 0.05$ represents statistical significance. The gut microbiota and metabolomic correlations were studied using Spearman's correlation analysis.

## Results

### General state of rats in each group

Prior to the experiment, all rats were generally in good condition, with shiny hair, sensitive reactions, and a normal diet. The rats in the three groups were kept in separate cages. After modeling and administration, the rats in the sham groups were well fed, with markedly increased weight, usual activities, good appetite, and glossy, smooth fur. In contrast, food intake, body weight, and gain in body weight were significantly lower in the 5/6 Nx groups compared with the sham group, and rats exhibited dull and yellow fur. After 12 weeks of CUR administration, compared with the 5/6Nx group, the rats in the administration group showed improvements in their mental state and appetite. Body weight changes of the three groups of rats during the experiment are shown in Table 1. All animals in the sham and the treatment groups survived till the end of the study, whereas two animals died in the 5/6Nx group.

### CUR improved kidney function and ameliorated fibrosis in 5/6 nephrectomy rats

To assess whether CUR could ameliorate renal injury, we used the classic 5/6 nephrectomy rat model, which is an ideal animal model for the study of chronic renal failure (CRF) and renal fibrosis. Scr, BUN, and 24H-P are the most used indicators of kidney function for routine clinical use. In Fig 2A, we can see that in contrast with the sham group, BUN, Scr, and 24-hour urine protein content in the 5/6Nx group were obviously increased. Following treatment with CUR, BUN, Scr, and 24H-P levels were markedly decreased compared with those detected in the 5/6Nx group.

Additionally, the results of Sirius red and Masson's trichrome staining demonstrated that CUR alleviated the pathological progression of CKD. Fig 2B and 2C displayed that the area of blue collagen fibers was less than that of the 5/6Nx group. Accumulation of ECM and tubular injury were also ameliorated compared with that in the 5/6Nx group (Fig 2B and 2D). Fibronectin (*FN*), *ACTA2* and *COL1A1*are the marker genes of renal fibrosis. RT-PCR analysis showed that 5/6 Nx upregulated the expression of *FN*, *ACTA2* and *COL1A1*, while CUR decreased the genes levels of these markers (Fig 2E). TGF-β/Smads pathway plays a key role in the development of renal fibrosis. As shown in Fig 2F and 2G, the expression of TGF-β1 in the 5/6 Nx group was apparently increased than in the sham group. Treatment with CUR significantly inhibited the expression of TGF-β1. In line with the above result, the positive regulatory factor Smad2 and Smad3 of the TGF-β signaling pathway in the 5/6 Nx group was significantly up-regulated, which was ameliorated by the administration of CUR.

**Table 1. Effect of CUR on physiological parameters of rats with CKD induced by 5/6Nx.**

|  | Sham | 5/6Nx | 5/6Nx+CUR |
|---|---|---|---|
| Initial body weight (g) | 200.49±10.57 | 200.5±9.47 | 201.22±9.39 |
| Final body weight (g) | 274.79±12.08 | 151.28±7.52** | 180.87±9.14## |
| Change in body weight (%) | 27.05±1.71 | -32.57±2.05** | -11.27±1.01## |

Data were presented as mean ± standard error.

Nx:Nephrectomy, CUR:curcumin.

*: compared between the sham and 5/6Nx group

**$P < 0.01$

#: compared between the 5/6Nx group and 5/6Nx+CUR groups

##$P < 0.01$.

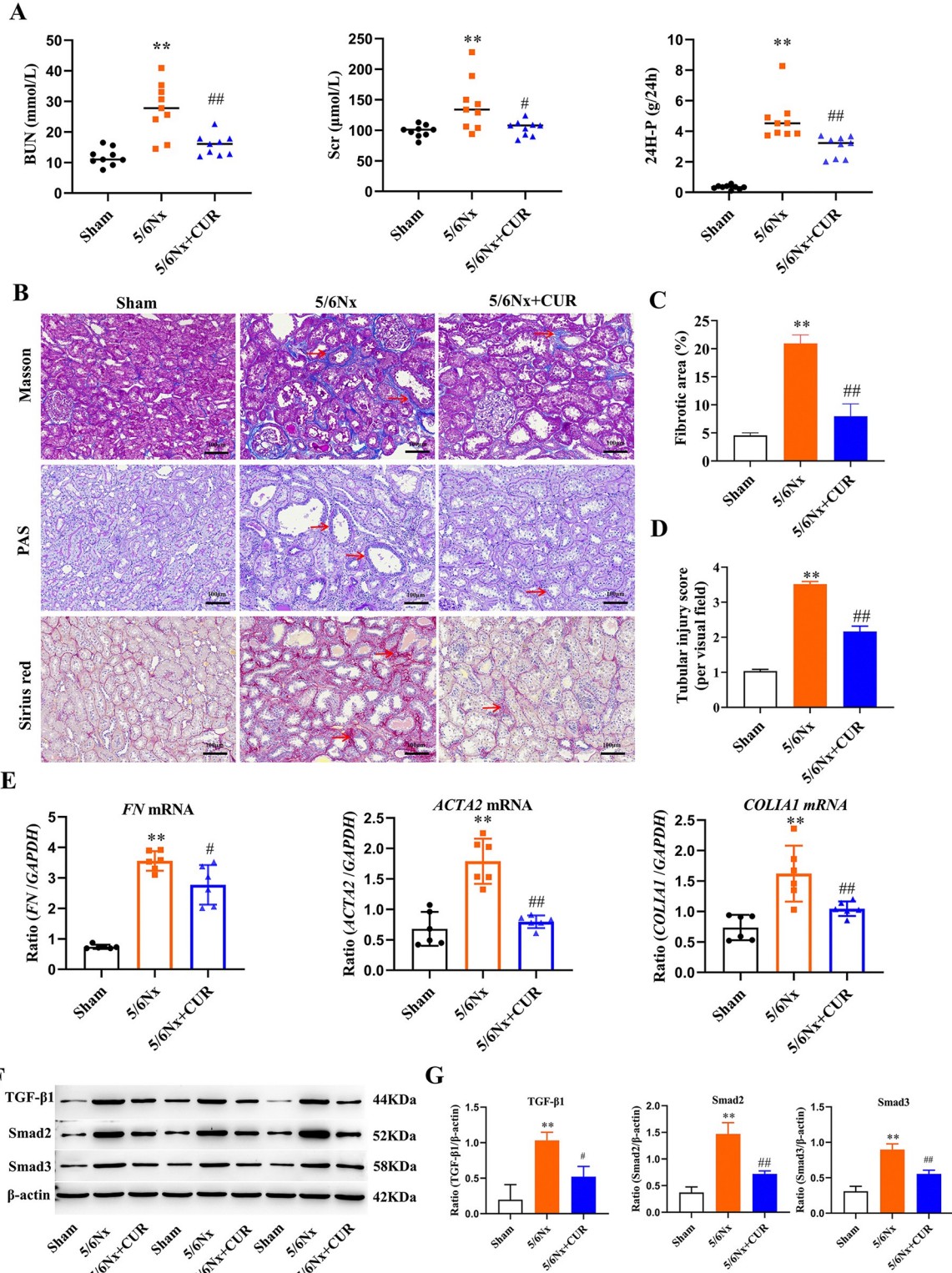

**Fig 2. CUR ameliorated kidney function and fibrosis in 5/6 Nx rats.** (A) The serum of BUN, Scr, and 24H-P levels were detected by commercial kits. (B) Change of Kidney pathology: representative image of Masson, PAS, and Sirius red staining of renal sections (Masson and PAS, scale bar = 100 μm; Sirius red, scale bar = 100 μm). (C) Quantitative analysis of fibrotic area based on Masson staining. (D) Quantitative analysis of tubular injury based on PAS staining. (E) The levels of *FN*, *ACTA2*, and *COL1A1* were determined by RT-PCR in the kidney. (F) Kidney expression of TGF-β/Smad signaling pathway assayed by Western blot in different groups. (G)

Quantitative analysis of Fig 2F. **$P$ <0.01, Compared with shams; #$P$ < 0.05, ##$P$ < 0.01, compared with 5/6Nx+CUR group, NS means no significant difference. Nx: Nephrectomy, CUR: curcumin, BUN: urea nitrogen, Scr: Serum creatinine, 24H-P: 24h urine protein quantitation, PAS: periodic acid-Schiff, *FN*: fibronectin, *ACTA2*: alpha-SMA, *COL1A1*: Collagen I, RT-PCR: Reverse transcriptase-polymerase chain reaction, TGF-β: Transforming growth factor-β.

## CUR inhibited inflammation in 5/6 nephrectomy rats through TLR4/NF-κB signaling pathway

Our next step was to investigate the state of inflammation in the kidneys. The kidney and serum levels of LPS were assessed in the sham, 5/6Nx, and 5/6Nx+CUR groups. Compared with the sham group, LPS in serum and kidney homogenate of 5/6Nx rats was significantly increased. CUR treatment significantly reduced the serum and kidney LPS accumulation in 5/6Nx rats (Fig 3A). Moreover, CUR suppressed abnormal expression of inflammatory cytokines by decreasing the mRNA expressions of *IL−1β*, *IL-6*, and *TNF-α* in the kidney (Fig 3B). Similarly, renal MCP-1 levels measured by ELISA and its mRNA expression were also markedly inhibited after CUR treatment (Fig 3C and 3D).

A study of the expression of the TLR4/NF-κB signaling pathway, which plays a critical role in renal injury and inflammation, was conducted in an effort to discover how CUR reduces inflammatory cell infiltration. Western blot analysis showed that TLR-4, p-IκB-α, IκB-α, p-NF-κB p65, and NF- κB p65 were elevated in 5/6Nx rats, and the expression of TLR-4, p-IκBα/IκBα and p-NF-κB p65/NF-κB p65 were significantly inhibited by CUR (Fig 3E and 3F). The results indicated that CUR inhibited TLR4/NF-κB signaling pathways recruited by LPS.

## CUR improved intestinal barrier function in 5/6 nephrectomy rats

Impaired intestinal mucosal barrier function can lead to increased intestinal permeability and translocation of bacterial endotoxins and thus exacerbate the development of CKD. Therefore, the influence of CUR treatment on intestinal barrier integrity was studied. H&E and PAS staining sections of the ileum showed that a small amount of intestinal gland structure was disrupted, the normal structure of the villi disappeared, and villi lodging or shedding was observed in rats from the 5/6 Nx group. Expectedly, CUR treatment significantly improved the structure of the ileum tissue (Fig 4A and 4B). After CUR intervention, the glands were arranged neatly in the colon, and the goblet cells were similar to those of the sham group (Fig 4C). PAS staining demonstrated that CUR effectively increased the number of goblet cells (Fig 4D). The pathological scores in the histological sections of the ileum and colon were shown in Fig 4E and 4F. Moreover, CUR therapeutic significantly reduced the secretion of inflammatory cytokines (*TNF-α*, *IL-6*, and *IL-1β*) and enhanced the expression of *Oclin*, *Cldn1*, and *ZO-1* mRNA and protein in both the ileum and colon (Fig 4G–4L).

## CUR altered the structural composition of gut microbiota in 5/6 nephrectomy rats

Next, we used 16S rDNA sequencing to reveal the impact of CUR on CKD-associated intestinal flora. Flower plots (samples >5) were mainly used to display the distribution of unique and common ASV (Amplicon Sequence Variant) in each sample, as shown in Fig 5A, ASV counts ranged from 414–668 bp, and 19 ASVs were common to all fecal samples. The dilution curves of all samples tended to be flat, indicating that the sample size in this experiment was reasonable and the species composition richness was high (Fig 5B). The administration of CUR increased observed features, Chao 1 richness, and Shannon, Simpson diversity, although these changes were not statistically significant (Fig 5C). Principal component analysis (PCA)

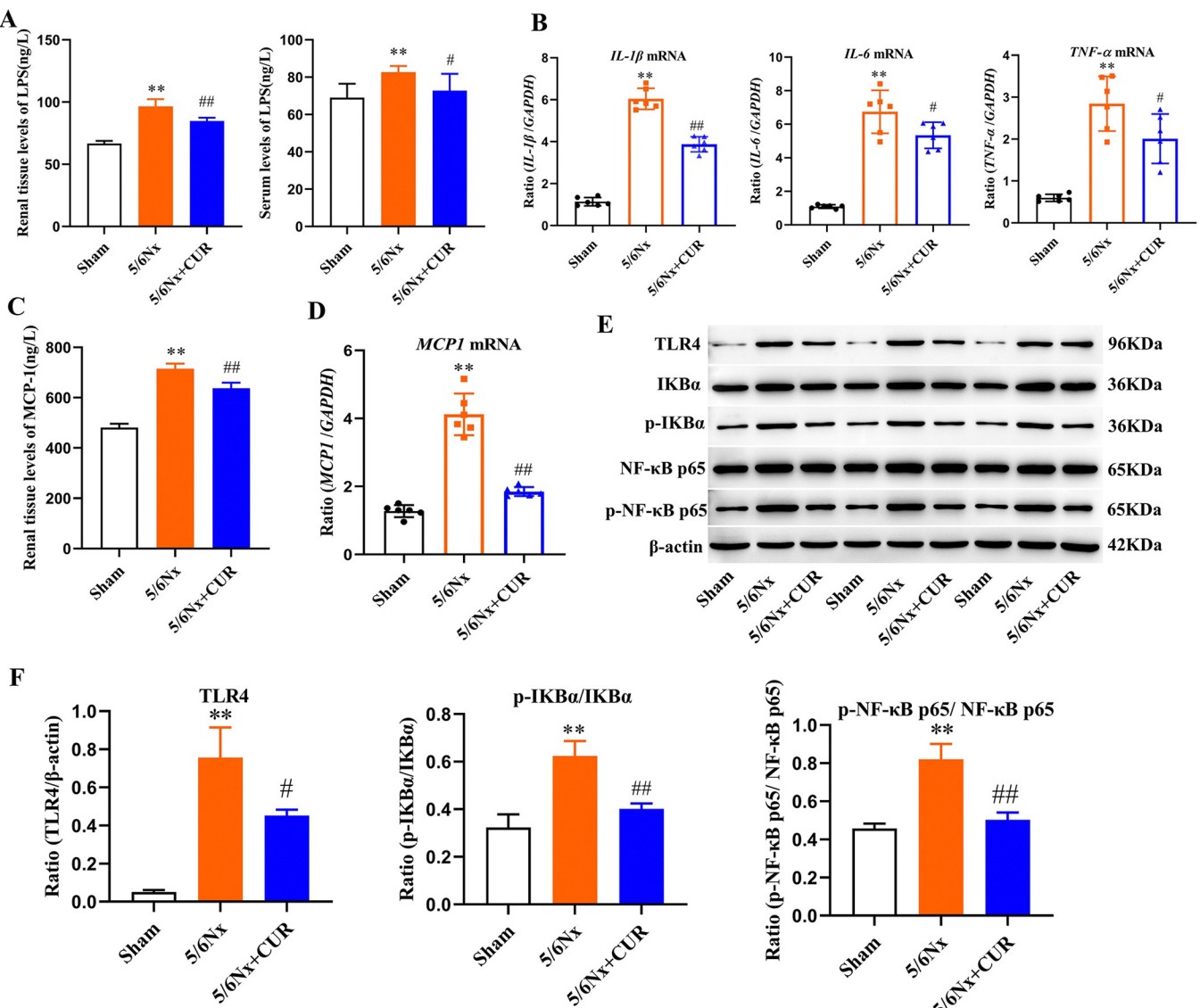

**Fig 3. CUR inhibited inflammation in 5/6 Nx rats.** (A) The serum and renal LPS levels were measured by ELISA. (B) The levels of *IL-1β*, *IL-6*, and *TNF-α* by RT-PCR in the kidney. (C) The renal MCP-1 levels were measured by ELISA. (D)The levels of *MCP-1* by RT-PCR in kidney tissues. (E) Kidney expression of TLR-4, p-IκB-α, IκB-α, p-NF-κB p65, and NF-κB p65 displayed by Western Blot strip chart. (F) Quantitative analysis of Fig 3E. **$P <0.01$, compared with the shams; #$P < 0.05$, ##$P < 0.01$, compared with 5/6Nx+CUR group. Nx: Nephrectomy, CUR: curcumin, LPS: Lipopolysaccharides, ELISA: enzyme linked immunosorbent assay, *IL−1β*: interleukin −1β, *IL-6*: interleukin-6, *TNF-α*: tumor necrosis factor-α, MCP-1: Monocyte chemoattractant protein-1,RT-PCR: Reverse transcriptase-polymerase chain reaction, TLR4: Toll-like receptor 4, NF-κB p65: Nuclear transcription factor-kappa B p65, p-NF-κB p65: Phospho-Nuclear transcription factor-kappa B p65.

and Principal Co-ordinates Analysis (PCoA) showed differences between the three groups. After CUR treatment, the https://fanyi.baidu.com/ - ##species composition of gut microbiota was restructured, and showed a callback trend to the sham group (Fig 5D and 5E).

At the phylum level, *Firmicutes* were the main phyla of microbiota in the three groups, followed by *Bacteroidota*. In contrast with the 5/6Nx group, CUR significantly increased the relative abundance of *Bacteroidota* and reduced the ratio of *Firmicutes/ Bacteroidota* (Fig 5F). At the order level, compared with shams, the relative abundance of *Erysipelotrichales* and *Sphingomonadales* was obviously higher in the 5/6Nx group, while *Bacteroidales* were less abundant.

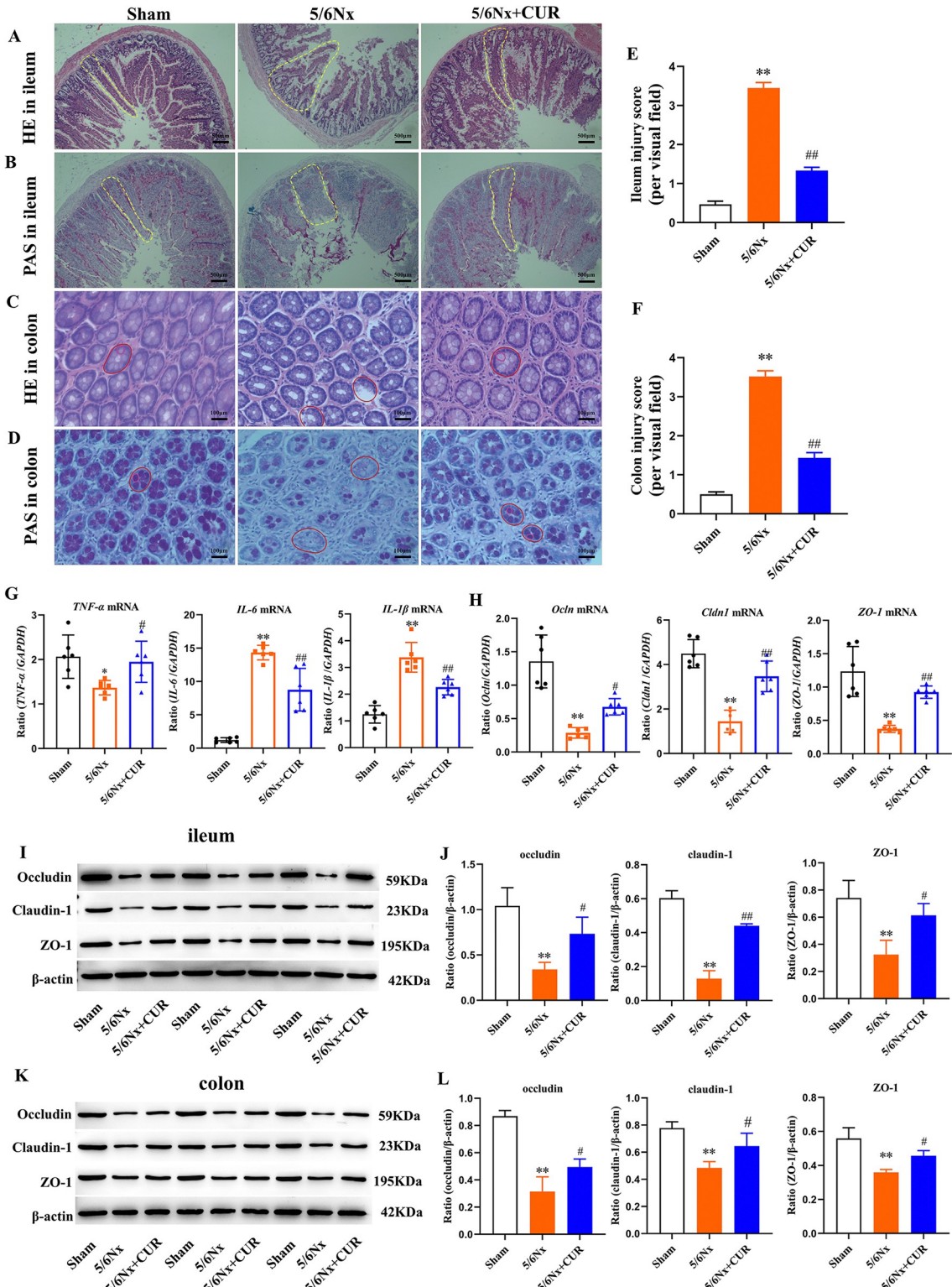

**Fig 4. CUR improved intestinal barrier function in 5/6 nephrectomy rats.** (A, B) Representative image of H&E and PAS staining of ileum sections (scale bar = 500 μm). (C, D) Representative image of H&E and PAS staining of colon sections (scale bar = 100 μm). (E, F) ileum and colon histopathology score. (G)The levels of inflammatory factors by RT-PCR in the ileum. (H) The levels of *Ocln*, *Cldn1*, and *ZO-1* by RT-PCR in the ileum. (I, J) Western blot analysis of occludin, claudin-1, and ZO-1 protein in ileum tissues. (K, L) Western blot analysis of occludin, claudin-1 and ZO-1 protein in colon tissues.*P <0.05, **P <0.01, compared with the shams;

$^{#}P < 0.05, ^{##}P < 0.01$, compared with the 5/6Nx+CUR group.Nx: Nephrectomy, CUR: curcumin, H&E: hematoxylin-eosin, PAS: periodic acid-Schiff, *IL−1β*: interleukin −1β,*IL-6*: interleukin-6, *TNF-α*: tumor necrosis factor-α, *Ocln*: Occludin; *Cldn1*: claudin-1, *ZO-1*: zonula occludens-1,RT-PCR: Reverse transcriptase-polymerase chain reaction.

Administration of CUR decreased the relative abundance of *Erysipelotrichales* and *Sphingoonadales* genera and enriched the relative abundance of *Bacteroidales* (Fig 5G). At the genus level, *Muribaculaceae*, which decreased in the 5/6 Nx group, showed more relative abundance in the 5/6 Nx+CUR group. CUR treatment also enriched SCFA-producing bacteria *Lachnospiraceae_NK4A136_group* and *Eubacterium_siraeum_group* significantly. Compared with the 5/6Nx group, *NK4A214_group* and *Family_XIII_AD3011_group* showed a decreasing trend in the 5/6 Nx+CUR group (Fig 5H–5J). These results suggest that 5/6Nx reduced the relative abundance of beneficial bacteria, and CUR treatment reversed this trend. Therefore, we speculate that CUR has a beneficial effect on reshaping the gut microbiota, increasing the abundance of beneficial bacteria, and potentially offering beneficial effects.

## CUR significantly improved gut microbial function associated with fibrosis

To investigate the role of the gut microbiota in our study, PICRUSt2 analysis was utilized to predict the function of the bacterial microbiota. The random forest model was used to distinguish the gut microbiome of different genera among the three groups, and different genera were ranked based on their contribution to the total gut microbiome. As shown in Fig 6A, *Muribaculaceae*, *Eubacterium_siraeum_group*, and *Lachnospiraceae_NK4A136_group* all ranked in the front, which further proved that the above three bacteria genera may have played an important role in the prevention of CKD by the CUR, although evidence was insufficient. Additionally, the KEGG pathway and COG analysis were used to explore potential differences in the functional composition of the microbiome between 5/6 and 5/6Nx+CUR. Functional predictions indicated that bacterial colony function was associated with metabolic pathways, particularly related to amino acid metabolism, sulfur relay system, ribosome metabolism, nicotinate, nicotinamide metabolism, terpenoid backbone, and biosynthesis (Fig 6B and 6C). Spearman's correlation analysis demonstrated a significant negative correlation between *Muribaculaceae*, *Eubacterium_ siraeum_ group*, *Lachnospiraceae_NK4A136_group* and CKD-related biochemical indicators, including BUN, 24H/P, key proteins of the TLR4 pathway and LPS, indicators of fibrosis and inflammatory factors. This indicated that these bacteria may have potential anti-renal fibrosis effects. https://fanyi.baidu.com/ - ##javascript:void(0);Conversely, *NK4A214_group* and *Family_XIII_AD3011_group* displayed a strong positive correlation with the aforementioned indicators, indicating that CKD development was associated with an increase in the abundance of these gut microorganisms (Fig 6D). Our results suggest a link between renal fibrosis and an increased abundance of these specific gut microbiota.

## CUR increased the levels of acetate and Vitamin D2, as well as regulated the function of metabolic pathways in 5/6Nx rats

Utilizing untargeted LC-MS metabolomics research techniques, we investigated the changes in plasma metabolomics in response to the treatment of CUR. We used unsupervised PCA to analyze the metabolic profiles of all rat plasma samples, and it could be observed that the samples from the 5/6Nx and 5/6 Nx+CUR groups were well distinguished from those of the shams (Fig 7A). A supervised Orthogonal projection to latent structure-discriminant analysis (OPLS-DA) was carried out to reveal the differences in metabolic profile between the three groups. As shown in Fig 7B, the 5/6 Nx group and 5/6 Nx+CUR group were two different

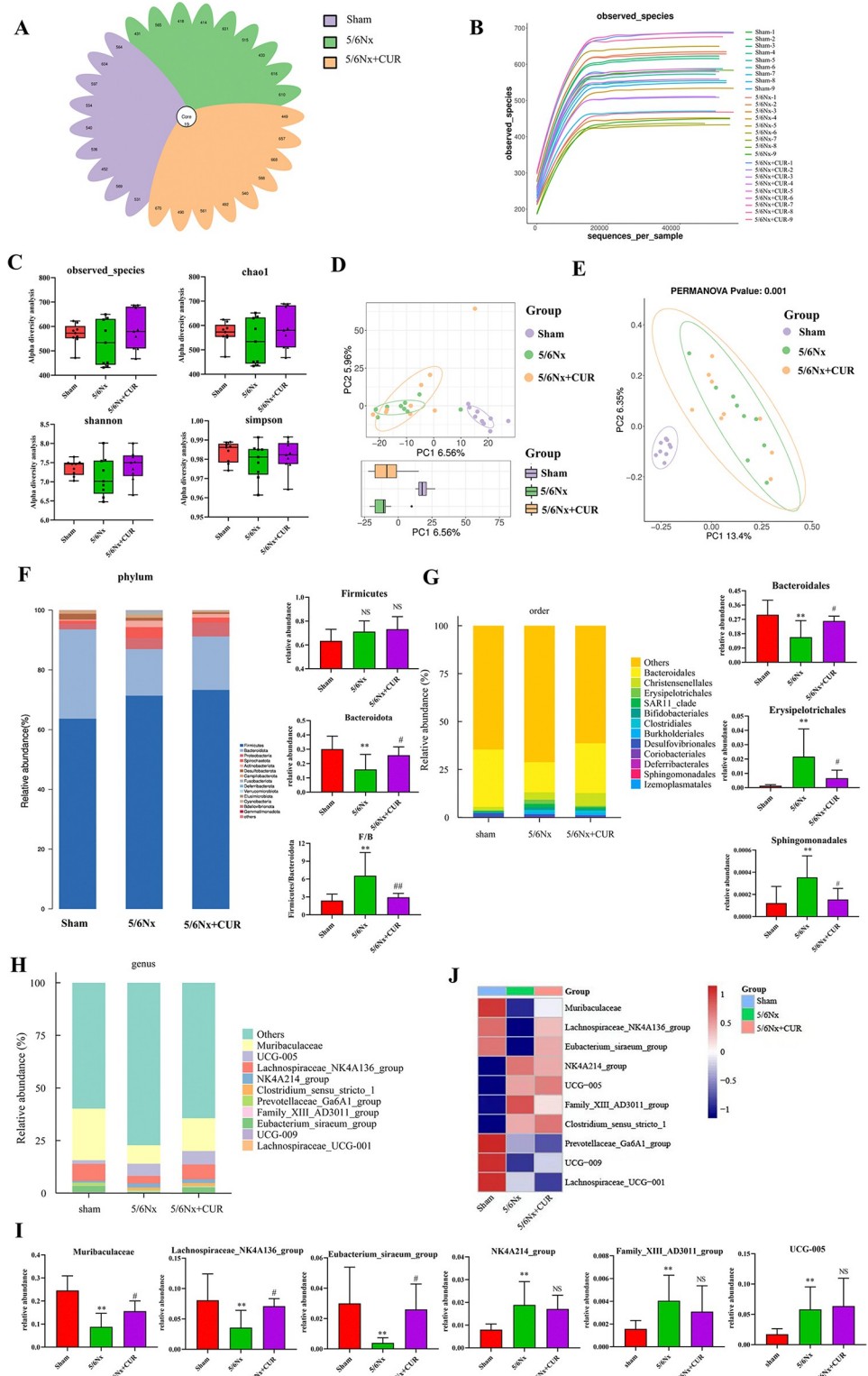

**Fig 5. CUR altered the structural composition of gut microbiota in 5/6 nephrectomy rats.** (A) Flower plots diagram. (B) Rarefaction curve measuring the amount of sequencing data. (C) Observed features, Chao 1, Shannon, and Simpson indexes in α-diversity analysis. (D, E) Analysis of β-diversity of gut microbiome in different groups. PCA of gut microbiota between all groups (D), PCoA between all groups(E). (F) The species composition at the phylum level. (G) The species composition at the order level. (H, I) The species composition and relative abundance at the

genus level. (J) Heat maps of the relative abundance of species composition at the genus level. $*P < 0.05$, $**P < 0.01$, compared with shams; $^{#}P < 0.05$, $^{##}P < 0.01$, compared with the 5/6Nx+CUR group, NS means no significant difference. Nx: Nephrectomy, CUR: curcumin, PCA: Principal Component Analysis, PCoA: Principal Co-ordinates Analysis.

clusterings, suggesting that the metabolic patterns of the 5/6 Nx+CUR group were obviously different from the 5/6 Nx group. The permutation test (n = 200) indicated that all OPLS-DA models did not overfit and had good explanatory and predictive abilities (Fig 7C).

A total of 1911 variables were selected based on the S-Plots (Fig 7D), using VIP ≥1 and P-value<0.05 as screening criteria. A total of 343 potential differential metabolites were identified between the sham group and the 5/6Nx group (Fig 7E, S2 Table). CUR restored the changes of 25 significant differential metabolites in CKD rats, as shown in Fig 7F. CUR treatment up-regulated content of acetate and vitamin D derivatives, while it down-regulated a variety of uremic retention solutes (URS), such as Tryptophan 2-C-mannoside and 1-Methylhypoxanthine (Fig 7G). Meanwhile, the multivariate ROC analysis based on the 3 metabolites showed higher AUC values: acetate, 0.975; Vitamin D, 0.889; Tryptophan 2-C-mannoside, 0.79 (Fig 7H). Furthermore, the screened differential metabolites were subjected to KEGG metabolic pathway analysis. The differential metabolites in 5/6Nx vs 5/6Nx+CUR groups were mainly enriched in Glycerophospholipid metabolism and Fat digestion and absorption (Fig 7I). Further analysis showed that the down-regulated metabolites were mainly enriched in glycine, serine, threonine, and glycerophospholipid metabolism (Fig 7J). The up-regulated metabolites were mainly concentrated in fat digestion and absorption, cAMP signaling pathway, phosphatidyl inositol signaling system, and sulfur metabolism (Fig 7K). It is speculated that the above pathway may be the key pathway for CUR to exert an anti-renal fibrosis effect through microbiota-derived metabolites.

## The potential relationship between metabolites and gut microbiota, as well as the correlation with CKD-related biochemical indicators

To better understand the relationship between altered gut microbiota and differential serum metabolites, the top 10 species at the genus level and the 25 differential serum metabolites based on Pearson's correlation coefficients were analyzed (Fig 8A). The contents of Vitamin D2 and acetate improved in the 5/6Nx+CUR group, and the above two metabolites were significantly positively correlated with *Lachnospiraceae_NK4A136_group* and *Eubacterium_siraeum_group*. However, the metabolites declined after administering CUR, including 1-Methylhypoxanthine and Tryptophan 2-C-mannoside, and were negatively correlated with the bacteria as mentioned above, whereas positively correlated with *Lachnospiraceae_UCG-001*. The correlations between metabolites and CKD-related biochemical indicators were also evaluated (Fig 8B). Plasma Tryptophan 2-C-mannoside and 1-Methylhydroxyxanthine were significantly positively correlated with BUN, 24H/P, and indicators of fibrosis and inflammation, such as fibrosis index, Collagen I, FN, and LPS, TLR4 to induce MyD88/NF-kappaB signaling. Reciprocally, acetate, a short-chain fatty acid, was reported to have an anti-fibrosis effect. Our results showed that plasma acetate negatively correlated with the abovementioned indicators. Overall, CUR treatment effectively enriched the abundance of the beneficial bacteria, such as *Lachnospiraceae_NK4A136_* and *Eubacterium_siraeum_group*, along with increases in plasma acetate concentrations in 5/6Nx rats.

## Discussion

In this study, we determine the effect of CUR in the treatment of CKD, gut microbiota dysbiosis, and related metabolites that may contribute to renal fibrosis. Specifically, it has been

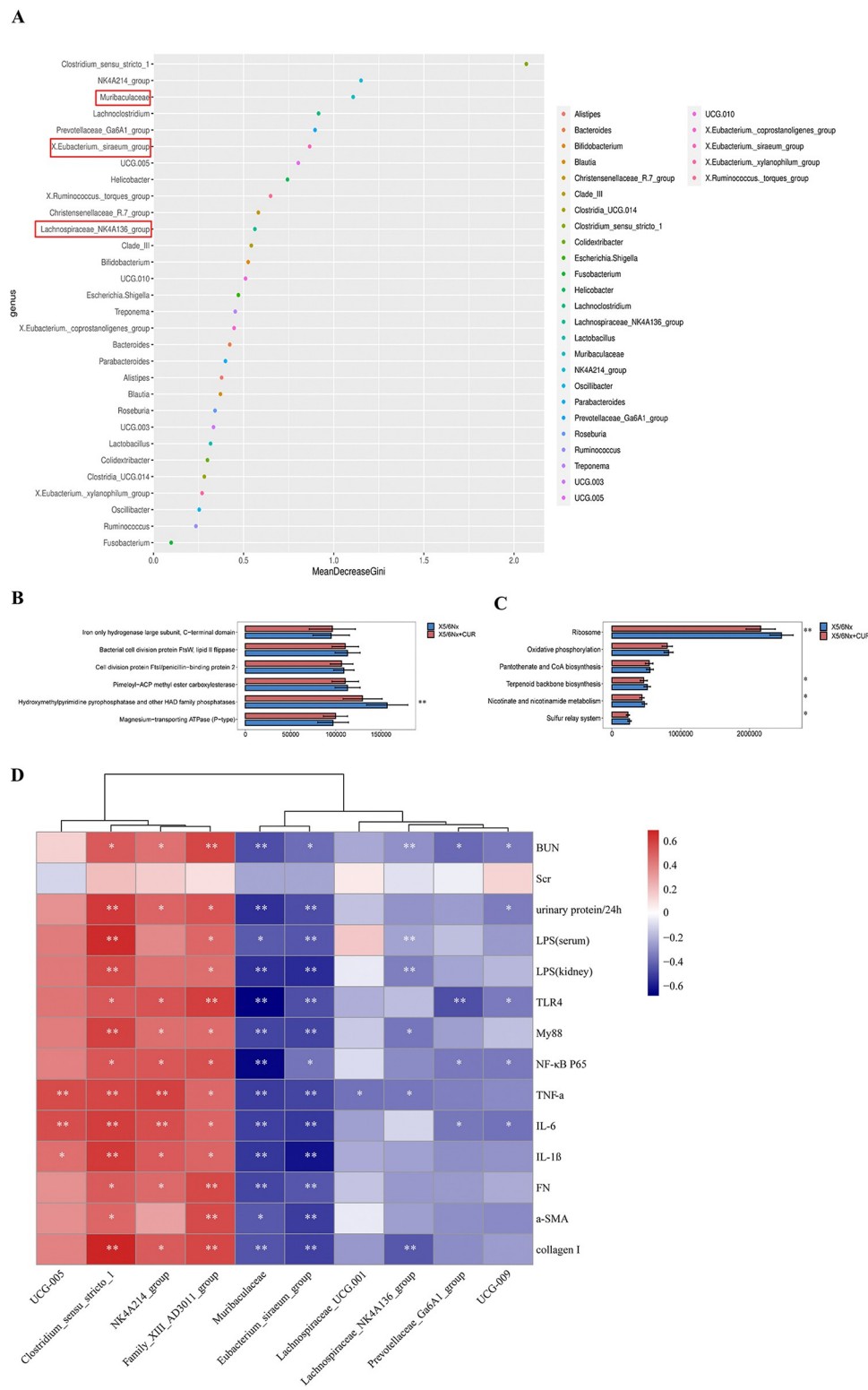

**Fig 6. Functional analysis of the microbial community.** (A) Random forest map analysis. (B) Contrast of COG categories data and (C) KEGG pathway results between 5/6Nx and 5/6Nx +CUR group using PICRUST. (D) Correlation analysis of gut microbiota and CKD-related biochemical indicators. *$P$ <0.05, **$P$ < 0.01, compared to the 5/6Nx group. Nx:Nephrectomy, CUR: curcumin, COG: a cluster of orthologous groups, KEGG: Kyoto Encyclopedia of Genes and Genomes.

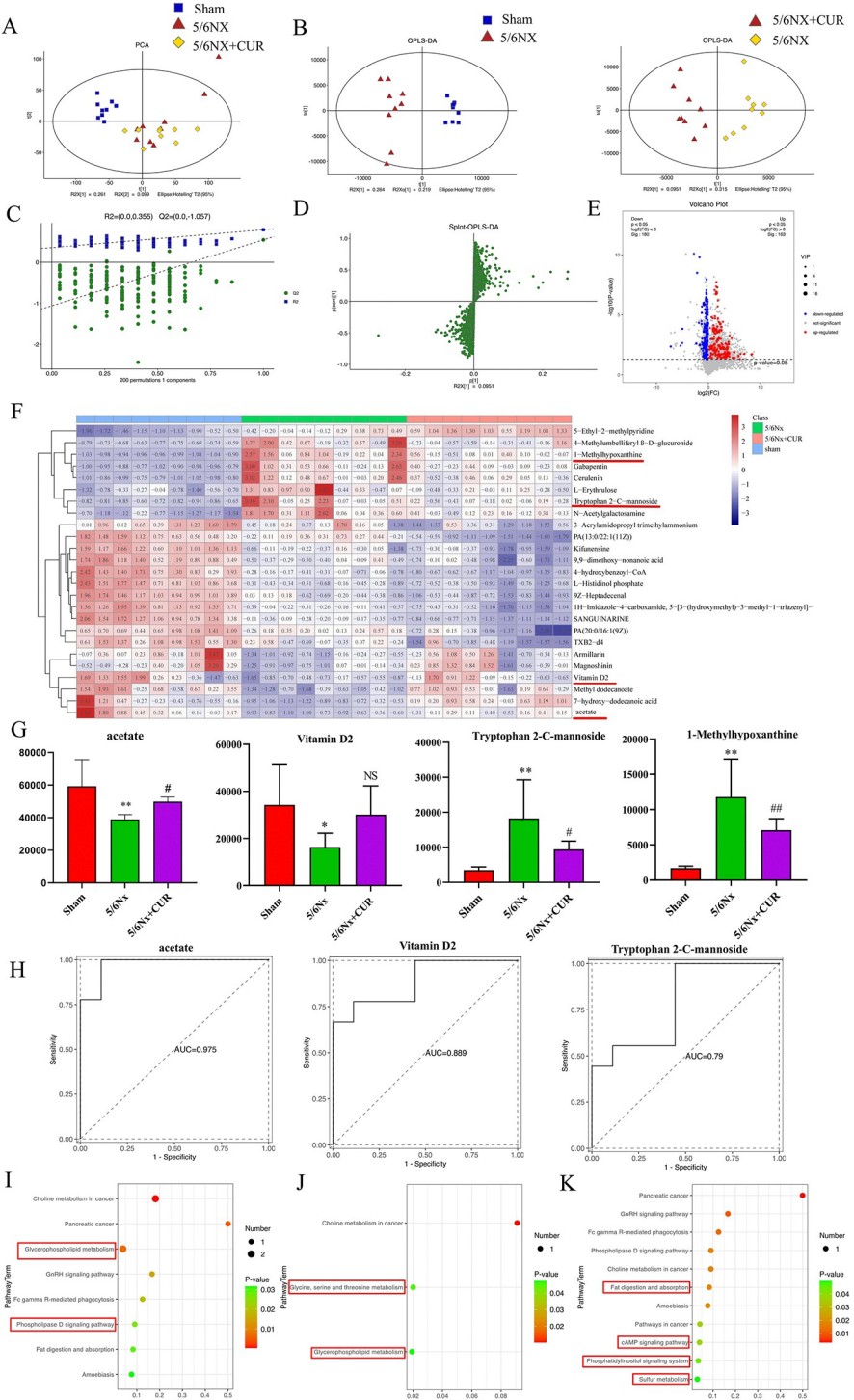

**Fig 7. Protective effects of CUR administration on metabolites derived from microbiota.** (A) PCA analysis in different groups. (B) OPLS-DA analysis between sham and 5/6Nx groups (left) and 5/6Nx and 5/6Nx+CUR groups (right). (C) The permutation test for the OPLS-DA model was generated from the serum profile of all samples. (D) The Splot-OPLS-DA analysis of the sham and 5/6Nx group. (E) Volcanic maps of metabolites in sham and 5/6Nx group. (F-G) Relative abundance of representative differential serum metabolites in different groups. (H) Three metabolites in the 5/6Nx and 5/6Nx+CUR groups were analyzed based on PLS-DA ROC curves, associated AUC, and 95% CI. (I) Summary of KEGG pathway analysis of 5/6Nx vs 5/6Nx+CUR. (J) CUR down-regulated differential metabolite KEGG pathway. (K) CUR up-regulated differential metabolite KEGG pathway. *$P < 0.05$, **$P < 0.01$, compared with shams;

#$P < 0.05$, ##$P < 0.01$, compared with the 5/6Nx+CUR group; NS means no significant difference. Nx:Nephrectomy, CUR: curcumin, PCA: principal component analysis, OPLS-DA: Orthogonal projection to latent structure-discriminant analysis, PLS-DA: Partial Least Squares Discriminant Analysis, ROC: Receiver Operation Characteristic, AUC: Area Under the Curve, CI: Confidence Interval, KEGG: Kyoto Encyclopedia of Genes and Genomes.

demonstrated that CUR could effectively slow the fibrosis progression of CKD induced by 5/6 Nx, including improving renal pathological changes and renal function. CUR down-regulated renal fibrosis parameters (FN, a-SMA, and collagen I), which might indicate an inhibition of the TGF-β1/Smads signaling pathway. Next, we showed that CUR attenuated intrarenal inflammation by preventing the TLR4/NF-κB signaling pathway activity. LPS, an important component of the outer membrane of Gram-negative bacteria, is the ligand of TLR4 [15, 23]. LPS interaction with TLR4 activates NF-κB signaling pathways, which mediates the secretion of a large number of inflammatory cytokines (TNF-α, IL-6, IL-1β) and chemotactic proteins, such as MCP-1, to strengthen the inflammatory response of the kidney and form an "inflammatory cascade" that finally results in renal fibrosis.

The intestinal mechanical barrier is an important structure for maintaining the function of the intestinal epithelium, and tight junction proteins are important constituents of the mechanical barrier, which https://fanyi.baidu.com/ - ##javascript:void(0);occupy an important role in regulating intestinal permeability [24, 25]. Interestingly, some studies showed that the kidney damage caused by intestinal endotoxins was associated with intestinal mucosal barrier injury [26, 27]. Meanwhile, it has been reported that CUR could restore tight junction protein expression, including ZO-1 and occludin, and enhance intestinal barrier integrity in metabolic disease [28]. In the present study, CUR treatment improved the morphology of the ileum and colon, upregulated ileal occludin, claudin-1, and ZO1 expression, and lowered levels of inflammatory cytokines release. Thus, we speculate that CUR treatment could prevent the translocation of gut-derived endotoxin LPS from intestinal leakage by restoring intestinal barrier function.

Gut microbiota, a complex microecosystem and one of the largest immune organs in the human body, may contribute to the occurrence and development of CKD [29]. Multiple clinical studies have shown differences in gut microbiota structure between CKD patients and healthy people [30, 31]. Meanwhile, dysregulation of gut microbiota was also observed in rodent models of CKD [32]. In this study, high-throughput sequencing results showed that the diversity and species richness of gut microbiota were altered in CKD rats, which were reversed by CUR. At the phylum level, the increase in the *Firmicutes/Bacteroidota (F/B)* ratio is considered a marker of gut microbiota disorder. Previous studies have shown that dysregulation of gut microbiota could significantly increase the level of pro-inflammatory cytokines IL-6 and *F/B* ratio, suggesting a close relationship to the occurrence of inflammation [33]. Our study showed that CUR reduced the F/B ratio, indicating that CUR could improve intestinal flora disorder in CKD rats and inhibit inflammatory reactions. At the genus level, we detected an enrichment of *Muribaculaceae*, *Lachnospiraceae_NK4A136_group*, and *Eubacterium_siraeum_group* in 5/6Nx+CUR rats. *Muribaculaceae*, which belongs to Bacteroides S24-7, is the dominant family in the intestinal tract of rats and has been shown to improve intestinal barrier function and promote the increase of SCFA-producing bacteria [34, 35]. *Lachnospiraceae_NK4A136_group* and *Eubacterium_siraeum_group* are two important types of butyrate-producing bacterium. Hu et al., research has shown that a mouse model of unilateral ureteral obstruction (UUO) resulted in a lower relative abundance of *Lachnospiraceae_NK4A136_group* and appeared negatively correlated with fibrosis severity [36]. Recent research demonstrated that *Eubacterium* preferentially colonizes the mucous layer, thereby increasing

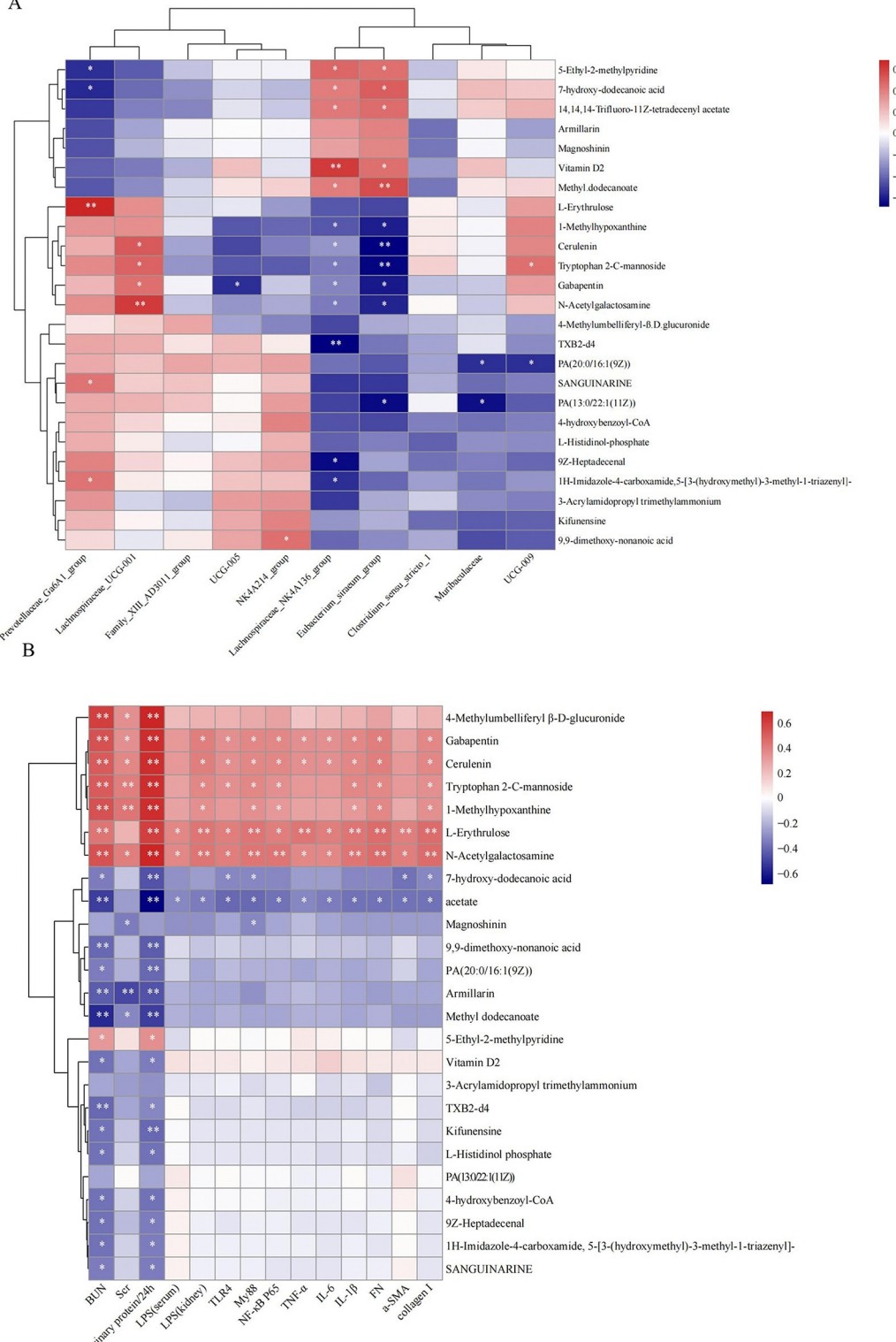

**Fig 8. Association analysis of differential metabolites with gut microbiota and physiological indices separately.** (A) Correlation analysis of plasma differential metabolites and gut microbiota at the genus level. (B) Correlation analysis of plasma differential metabolites and CKD-related biochemical indicators. Red represents a positive association, and blue represents a negative association. *P < 0.05, **P < 0.01. CKD: chronic kidney disease.

the bioavailability of butyrate for epithelial colon cells [37]. Our study indicated that 5/6 Nx decreased the relative abundance of beneficial bacteria, which was reversed by CUR treatment. Hence, we speculate that CUR positively affected renal fibrosis by restoring gut microbiota.

Furthermore, untargeted metabolomics was carried out to study the changes in serum metabolite levels and evaluate CUR's interventional effects in CKD rats. The metabolomics results revealed that after treatment with CUR, the levels of Vitamin D2 and acetate were upregulated, and the levels of Tryptophan 2-C-mannoside and 1-Methylhypoxanthine were downregulated. According to previous studies, both CKD and kidney transplant recipients were associated with vitamin D deficiency, which promotes renal fibrosis and functional impairment [38]. Multiple lines of evidence have shown that vitamin D and analogues decrease renal fibrosis progression via the vitamin D receptor and inhibit TGF-β1/Smad3 signaling pathway [39]. Acetate, a kind of SCFAs, He et al. have summarized SCFAs has multiple effects on CKD, such as reducing inflammatory response, inhibiting oxidative stress, regulating autophagy, regulating energy metabolism, and immune pathways [40]. Clinical research specifically targeted the CKD population shows that with the decrease of renal function, the reduction in the abundance of SCFAs-producing bacteria in the intestine of patients is further aggravated, and the degree of decline in the abundance of bacteria is positively correlated with the progression of CKD [3]. In an in vitro experiment, Xu et al. tested the influence of SCFA treatment on LPS-induced inflammation in HK-2 cells and RAW264.7 cells, suggesting SCFA treatment alleviated the inflammatory responses via inhibition of the iNOS and TLR4/NF-κB signaling pathway [41]. Tryptophan 2-C-mannosides belong to indole derivatives and participate in tryptophan metabolism regulated by gut microbiota directly or indirectly [42]. These intestinal metabolites, which are ligands for aryl hydrocarbon receptors, have been shown to participate in renal fibrosis [42]. 1-Methylhypoxanthine is a purine derivative belonging to the sub-class of purines and purine derivatives, which has a similar structure to adenosine and prevents adenosine receptors from acting pharmacologically by competitively inhibiting adenosine [43]. Jackson *et al.*, found a causal relationship between low adenosine levels and inflammation in patients with COVID-19-related acute kidney injury. A significant elevation in inflammatory cytokines was observed in COVID-19 patients with a poor renal prognosis and a reduction in extracellular anti-inflammatory purines [44]. The enrichment analysis of metabolites between the 5/6Nx group and the 5/6Nx+CUR group revealed that glycerophospholipid metabolism differed between the two groups. Glycerophospholipids are the most important in the kidney, acting as signaling molecules and anchors for a cell membrane protein, which were reportedly part of inflammation [45, 46]. As mentioned above, the earlier studies and the results of differential metabolites screened in our study suggested that CUR may have produced an anti-fibrotic effect in renal fibrosis via regulating multiple metabolites.

In addition, the relationships https://fanyi.baidu.com/ - ##between gut microbiota and metabolites were analyzed. The significant improvements in plasma acetate and Vitamin D levels in CKD rats after CUR treatment were linked to the increased relative abundances of *Lachnospiraceae_NK4A136_group* and *Eubacterium_siraeum_group*. However, through Spearman's analysis, we could not firmly conclude that these effective microbes directly contributed to the increased production of plasma acetate and Vitamin D. In the correlation analysis of metabolites and gut microbiota with the indicators of chronic kidney injury, respectively, plasma Tryptophan 2-C-mannoside and 1-Methylhydroxyxanthine may indicate their important roles in the occurrence of renal fibrosis. *Eubacterium_siraeum_group* and *Lachnospiraceae_NK4A136_group* were negatively correlated with indices of chronic kidney injury, and future research will focus on the mechanism of these beneficial microbiota against renal fibrosis.

There are some limitations of our study. Whether the https://fanyi.baidu.com/ - ##anti-renal fibrosis effects of CUR were dependent on gut microbiome was not directly confirmed, and further research using antibiotics, faecal microbiota transplantation, and probiotics will be needed. Additionally, only a correlation between key bacteria and differential metabolites in the study was carried out, and whether these differential metabolites were produced by gut microbiota remained uncertain. Finally, there are certain differences between the CKD rats model and CKD patients; developing high-quality randomized clinical trials with specific CKD populations will be necessary to confirm our results.

## Conclusion

In summary, our study showed that CUR effectively ameliorates kidney function and fibrosis in 5/6Nx rats by inhibiting TLR4/NF-κB and TGF-β1/Smads pathway. Additionally, CUR also restored the intestinal barrier and suppressed the levels of inflammatory factors. Based on an integrative analysis of gut microbiota and metabolomics results, CUR effectively remodeled the gut microbiota by increasing the levels of beneficial bacteria, including *Eubacterium_siraeum_group*, *Lachnospiraceae_NK4A136_group*, and *Muribaculaceae*. Furthermore, CUR also enhances Vitamin D2 and acetate production, which may have beneficial effects on the treatment of CKD. Specifically, we investigated the mechanism of action of CUR against renal fibrosis and found a strong correlation between CUR and the regulation of the gut microbiome, thereby suggesting that CUR inhibits renal fibrosis by affecting gut microbes.

## Supporting information

**S1 Table. Primer sets designed for quantitative real-time PCR.**
(DOCX)

**S2 Table. Differential metabolites between sham group and 5/6Nx group.**
(XLS)

**S1 File. Bioinformatics Analysis of 16S rDNA Sequencing and LC-MS metabolomic analysis technology.**
(DOC)

**S1 Raw images. The original uncropped and unadjusted images underlying all blot or gel data.**
(PDF)

## Acknowledgments

We express our sincere appreciation to Abid Naeem (Ph.D. from Key Laboratory of Modern Preparation of Traditional Chinese Medicine, Ministry of Education, Jiangxi University of Traditional Chinese Medicine) for his linguistic assistance during the preparation of this manuscript. We also sincerely appreciate the support of the "Xuncheng Talents" Youth cultivation Project of Jiujiang and the Youth Cultivation Project of Jiujiang University Affiliated Hospital.

## Author Contributions

**Conceptualization:** Xulong Chen, PuXun Tian.

**Data curation:** Cheng Li, Jingchun Yao, Meiren Li.

**Funding acquisition:** Xulong Chen.

**Investigation:** Weiwei Zha, Jiangwen Shen.

**Methodology:** Hongli Jiang.

**Supervision:** Xulong Chen.

**Writing – original draft:** Cheng Li, Jingchun Yao, Meiren Li.

**Writing – review & editing:** PuXun Tian.

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
