## [Decision Letter · Decision Letter 0]

11 Jul 2024

PONE-D-24-25478Curcumin modulated gut microbiota and alleviated renal fibrosis in 5/6 nephrectomy-induced chronic kidney disease ratsPLOS ONE

Dear Dr. Tian,

Thank you for submitting your manuscript to PLOS ONE. After careful consideration, we feel that it has merit but does not fully meet PLOS ONE’s publication criteria as it currently stands. Therefore, we invite you to submit a revised version of the manuscript that addresses the points raised during the review process.

The shortcomings of this paper needs to be worked out before it can be considered for publication. Therefore, we invite you to resubmit a revised version of the manuscript that addresses the points raised during the review process.

For your guidance, the reviewers' comments are included below.

Thank you for giving us the opportunity to consider your work.

We look forward to receiving your revised manuscript.

Kind regards,

Marianne Clemence, Staff Editor, on behalf of,

Palash Mandal 

Academic Editor

PLOS ONE

Journal Requirements:

"This project was assisted by National Natural Science Foundation of the People’s Republic of China (82270791), Jiangxi Traditional Chinese Medicine Science and Technology Plan Project (2022A137, 2023B1287), Jiangxi Provincial Health Commission Science and Technology Program (202212007), Science and Technology Research Project of Jiangxi Provincial Department of Education (GJJ201819, GJJ211805), Beijing Medical and Health Foundation (TYU046B) and Beijing Medical Award Foundation (YXJL-2022-0734-0294)."

Reviewers' comments:

Reviewer's Responses to Questions

**Comments to the Author**

1. Is the manuscript technically sound, and do the data support the conclusions?

Reviewer #1: Partly

Reviewer #2: Yes

2. Has the statistical analysis been performed appropriately and rigorously? 

Reviewer #1: No

Reviewer #2: Yes

3. Have the authors made all data underlying the findings in their manuscript fully available?

Reviewer #1: Yes

Reviewer #2: No

4. Is the manuscript presented in an intelligible fashion and written in standard English?

Reviewer #1: No

Reviewer #2: Yes

5. Review Comments to the Author

Reviewer #1: Reviewers' comments to editor:

This manuscript describes "Curcumin modulated gut microbiota and alleviated renal fibrosis in 5/6 nephrectomy-induced chronic kidney disease rats". The topic is of interest. However, there are several concerns about the study, and needs to be improved.

1- Why the authors did not add in vitro experiments to verify the in vivo results?

2-More details are needed on how the authors selected the dose and treatment period for , Curcumin? Please supplement the relevant references.

3- Why isn't there a separate curcumin group? Are additional effects of curcumin considered?

4- It is recommended to briefly describe the CKD model building method, and attach relevant references as in the article.

5- It is recommended to supplement the dilution ratio of relevant antibody indicators, and supplement the experimental procedure basis of important experiments, such as the source of the kit used in the pcr experiment.

6- It is recommended to increase the description of important details of 16S rDNA sequencing and non-targeted metabolome-related experiments, as current methods are too simplistic. The innovation of this paper lies in the application of multiple omics, but it is inappropriate for the author to put all the omics methods in the supporting materials..

7- The Figures in the manuscript is not clear enough and very fuzzy, it is recommended to revise according to the requirements of the magazine submission.

8- No arrows were used to mark specific pathological changes on all pathological pictures, Please add it.

9- You can see many groups in the protein results picture, but the article method does not seem to describe them in detail, please check.

Other comments：

1-It is suggested that supplement the recent three years of research references.

2-It is recommended to verify the appropriate selection and use of statistical methods. It is suggested to improve the selection basis and standard of statistical methods.

3-The flora should be written in italics, and the whole text should be revised uniformly

4- The article language should be carefully polished by a professional company or a native English speaker.

Reviewer #2: This manuscript eveals that curcumin has potential anti-renal fibrosis effects, and its mechanism may be related to its regulation of the composition of intestinal flora. This manuscript has good value for the application of CUR in anti-renal fibrosis. However, whether the anti-renal fibrosis effect of CUR depends on the regulatory effect on intestinal flora and the specific regulatory mechanism requires further experiments to verify. Since this study is closely related to metabolism, the author should provide the type of feedstuff, feeding status, body weight and survival rate of each group of rats in the animal experiment.

6. PLOS authors have the option to publish the peer review history of their article (what does this mean?). If published, this will include your full peer review and any attached files.

Reviewer #1: No

Reviewer #2: No

---

## [Author Response · Author response to Decision Letter 0]

16 Aug 2024

Dear editor,

Thank you very much for your letter regarding our manuscript entitled “Curcumin modulated gut microbiota and alleviated renal fibrosis in 5/6 nephrectomy-induced chronic kidney disease rats”. We also thank the reviewers’ constructive comments and suggestions. Our point-by-point response to the reviewers’ and suggestions are listed below again.

 We would like to submit the revised manuscript, and hope it is acceptable for publication in the journal.

Sincerely yours,

Cheng Li

Here are our responses to the reviewers’ comments one-by-one again. 

Journal Requirements:

Response:Thank you for your reminder.We have revised the manuscript according to the template you provided.

Response: Thank you for your valuable suggestion.(1) methods of sacrifice:in our study, we utilized isoflurane as an inhalant anesthetic, and the rats were subsequently euthanized. (2) methods of anesthesia and/or analgesia：anesthesia was induced with 4% isoflurane prior to surgery and maintained with 2.5% isoflurane throughout the procedure. (3) efforts to alleviate suffering:all experimental protocols adhered to the 3R principle (reduction, replacement, and refinement) to minimize pain and distress, with animals handled by trained personnel in a calm manner. This information has been incorporated into the Methods section of our manuscript.

Thank you for your comments.

"This project was assisted by National Natural Science Foundation of the People’s Republic of China (82270791), Jiangxi Traditional Chinese Medicine Science and Technology Plan Project (2022A137, 2023B1287), Jiangxi Provincial Health Commission Science and Technology Program (202212007), Science and Technology Research Project of Jiangxi Provincial Department of Education (GJJ201819, GJJ211805), Beijing Medical and Health Foundation (TYU046B) and Beijing Medical Award Foundation (YXJL-2022-0734-0294)."

Response: Thank you for your reminder. This work was supported by National Natural Science Foundation of the People’s Republic of China awarded to PuXun Tian (82270791) (https://kd.nsfc.cn/fundingProjectInit). Xulong Chen is supported by BELJING MEDICAL AWARD FOUNDATION (YXJL-2022-0734-0294) (https://www.yxjl.org/cms/html/index), BEUING MEDICAL AND HEALTH FOUNDATION (TYU046B) (http://www.ywjjh.org.cn/), Science Technology Research Project of Jiangxi Provincial Department of Education (GJJ211805), Jiangxi Traditional Chinese Medicine Science and Technology Plan Project (2023B1287) and Science and Technology Research Project of Jiangxi Provincial Department of Education (GJJ211805). Cheng Li received support from Jiangxi Traditional Chinese Medicine Science and Technology Plan Project (2022A137), Jiangxi Provincial Health Commission Science and Technology Program (202212007), Science and Technology Research Project of Jiangxi Provincial Department of Education (GJJ201819). The funders had no role in study design, data collection and analysis, decision to publish, and preparation of the manuscript. 

Thank you for your comments.

Response: Thank you for your reminder. All gut microbiome dataset files can be found online at https://www.ncbi.nlm.nih.gov/bioproject/PRJNA1069977

Additionally, four supplementary materials have been uploaded in the submission system.These four supplementary materials are as follows: S1 Table. Primer sets designed for quantitative real-time PCR. S2 Table. Differential metabolites between sham group and 5/6Nx group.S1 File. Bioinformatics Analysis of 16S rDNA Sequencing and LC-MS metabolomic analysis technology.S1 Raw images. The original uncropped and unadjusted images underlying all blot or gel data.

Thank you for your comments.

Response: Thank you for your suggestion.Our corresponding author has applied for a new ORCID iD (0000-0002-1408-9910).

Thank you for your comments.

Reviewers Requirements:

1- Why the authors did not add in vitro experiments to verify the in vivo results?

Response: Good question. The lack of in vitro experiments in this manuscript are mainly due to the following reasons:

1. Many in vitro studies[1-4] have shown that curcumin inhibits epithelial-mesenchymal transition (EMT) of rat tubular epithelial cells (NRK) or human HK-2 cells stimulated by TGF-β1. Therefore, conducting in vitro experiments on curcumin is not the focus of our study. 

2. Our study used fecal 16S rDNA gene sequencing to investigate the composition of gut microbiota in 5/6 nephrectomy rats. 16S sequencing is currently the sequencing method used in most articles on gut microbiota, but this sequencing method can only detect at the genus level and cannot be precise at the strain level. Therefore, it restricted our ability to extract strains for in vitro intervention on TGF-β1-induced NRK or human HK-2 cells. In subsequent experiment, we will use metagenomic sequencing to conduct in-depth exploration of bacterial strains.Although this study did not use metagenomic sequencing to screen strains for in vitro experiments, it still confirmed that curcumin reshape the composition of the gut microbiome as well as its metabolites in animal model of renal fibrosis, laying the foundation for future research in this direction.

3. Searching for articles published in journals of the same level in recent years, many articles related to gut microbiota do not have in vitro experimental data[5-8]. 

Thank you for your comments.

2-More details are needed on how the authors selected the dose and treatment period for , Curcumin? Please supplement the relevant references.

Response: Thank you for your valuable question. The dose of curcumin selected for this study is 200 mg/kg/d, and the treatment period is 12 weeks.Related references for dose selection of curcumin[9-13], relevant references for treatment period for curcumin[14-16].

Thank you for your comments.

3- Why isn't there a separate curcumin group? Are additional effects of curcumin considered?

Response: Good question. This study did not set up a blank administration group, mainly considering the following factors: 

1. The purpose of this study is to confirm that curcumin improves renal fibrosis in a 5/6Nx model; whether curcumin can reshape the intestinal flora and its metabolites in 5/6Nx rats and preliminary exploration of the relationship between gut bacteria and metabolites under the action of curcumin and renal fibrosis. Therefore, we sought to investigate the role of curcumin in a specific disease model, and no normal administration group was established.

2. As a safe natural metabolite, our study focuses on studying the role of curcumin in reshaping gut microbiota in 5/6 nephrectomy rats, rather than conducting toxicity experiments;

3. Referring to previous published literature, there are many articles on curcumin and gut microbiota in immune inflammatory diseases that do not add blank administration groups[17-19]. 

Thank you for your comments.

4- It is recommended to briefly describe the CKD model building method, and attach relevant references as in the article.

Response: Thank you for your question. I'm sorry for not providing a detailed description of the CKD model building method.CKD model construction is as follows:

Anesthesia was induced with inhaled 4% isoflurane before surgery and maintained with 2.5% isoflurane during surgery. The back of rats were depilated and prepared for surgery. Anesthetized rats were fixed in a prone position on a mouse board, the skin preparation area was disinfected with 75% alcohol, and sterile surgical drapes (self-made gauze) were placed. A longitudinal 2 cm surgical incision was made next to the spine at the lower left costal margin, the skin and subcutaneous tissue were cut layer by layer and the muscle was bluntly separated. Next, gently lifted the perirenal fat sac with forceps, placed the surgical handle on it, exposed and fixed the kidney outside the incision, peeled off the renal capsule and perirenal fat, exposed the upper and lower poles of the kidney, used a sharp blade to remove the upper and lower poles of the kidney, and quickly compressed the renal section with gelatin sponge for 30 seconds to stop bleeding, then returned the kidney and sutured layer by layer. One week later, the right kidney was exposed using the same method. After peeling off the renal capsule and perirenal fat, the 4-0 silk thread was used to ligate the renal pedicle along the renal hilum. Then, the right kidney was removed and sutured layer by layer after confirming no bleeding. The sham operation was performed only to peel the renal membrane without damaging the kidney. The other steps are the same as above. After operation, the rats were fed and watered freely in an SPF environment at room temperature and 45% relative humidity[20,21].

The above contents have been included in the methodology of the manuscript.Thank you for your comments. 

5- It is recommended to supplement the dilution ratio of relevant antibody indicators, and supplement the experimental procedure basis of important experiments, such as the source of the kit used in the pcr experiment.

Response: We sincerely thank the reviewer for the careful reading of our manuscript. According to the reviewer's suggestion, we have added the dilution ratio of relevant antibody indexes in the “western blot analysis” of materials and methods. The source information of the kit used in the PCR experiment was supplemented in “RNA extraction and Real-Time PCR” (S1 Table).

Thank you for your comments. 

6- It is recommended to increase the description of important details of 16S rDNA sequencing and non-targeted metabolome-related experiments, as current methods are too simplistic. The innovation of this paper lies in the application of multiple omics, but it is inappropriate for the author to put all the omics methods in the supporting materials.

Response: Thank you for your advice. We are sorry for not providing a detailed description of 16S sequencing and metabolomics methods in the method section. The description of important details of 16S rDNA sequencing and non-targeted metabolome-related experiments are as follows:

16S rDNA gene sequencing

Total genomic DNA was extracted using MagPure Soil DNA LQ Kit (Magan) following the manufacturer’s instructions. DNA concentration and integrity were measured with NanoDrop 2000 (Thermo Fisher Scientific, USA) and agarose gel electrophoresis. Extracted DNA was stored at -20°C until further processing. The extracted DNA was used as template for PCR amplification of bacterial 16S rRNA genes with the barcoded primers and Takara Ex Taq (Takara). For bacterial diversity analysis, V3-V4 (or V4-V5) variable regions of 16S rRNA genes was amplified with universal primers 343F (5’-TACGGRAGGCAGCAG-3’) and 798R (5’-AGGGTATCTAATCCT-3’) for V3-V4 regions. The Amplicon quality was visualized using agarose gel electrophoresis. The PCR products purified with AMPure XP beads (Agencourt) and amplified for another round of PCR. After purified with the AMPure XP beads again, the final amplicon was quantified using Qubit dsDNA Assay Kit (Thermo Fisher Scientific, USA). The concentrations were then adjusted for sequencing. Sequencing was performed on an Illumina NovaSeq 6000 with 250 bp paired-end reads. (Illumina Inc., San Diego, CA; OE biotech Company; Shanghai, China). Raw sequencing data were in FASTQ format. Paired-end reads were then preprocessed using cutadapt software to detect and cut off the adapter. After trimming, paired-end reads were filtering low quality sequences, denoised, merged and detected and cut off the chimera reads using DADA2 with the default parameters of QIIME2 (2020.11). At last, the software output the representative reads and the ASV abundance table. The representative read of each ASV was selected using QIIME2 package. All representative reads were annotated and blasted against Silva database Version 138 (or Unite) (16s/18s/ITS rDNA) using q2-feature-classifier with the default parameters.

LC/MS untargeted metabolomics analysis

80 μL of sample was added to a 1.5 mL Eppendorf tube with 10 μL of L-2-chlorophenylalanine (0.3 mg/mL) dissolved in methanol as internal standard, and the tube was vortexed for 10 s. Subsequently, 240 μL of ice-cold mixture of methanol and acetonitrile (2/1, vol/vol) was added, and the mixtures were vortexed for 1 min, and the whole samples were extracted by ultrasonic for 10 min in ice-water bath, stored at -20 ℃ for 30 min. The extract was centrifuged at at 4°C (13,000 rpm) for 10 min. 160 μL of supernatant in a glass vial was dried in a freeze concentration centrifugal dryer .240μL mixture of methanol and water (1/4, vol/vol) were added to each sample, samples vortexed for 30 s, extracted by ultrasonic for 3 min in ice-water bat, then placed at -20°C for 2 h. Samples were centrifuged at 4°C (13,000 rpm) for 10 min. The supernatants (150 μL) from each tube were collected using crystal syringes, filtered through 0.22 μm microfilters and transferred to LC vials. Finally, 15 μL supernatant was used for LC-MS detection.

The original LC-MS data were processed by software Progenesis QI V2.3 (Nonlinear, Dynamics, Newcastle, UK) for baseline filtering, peak identification, integral, retention time correction, peak alignment, and normalization. Main parameters of 5 ppm precursor tolerance, 10 ppm product tolerance, and 5% product ion threshold were applied. Compound identification was based on precise mass-to-charge ratio (M/z), secondary fragments, and isotopic distribution using The Human Metabolome Database (HMDB), Lipidmaps (

---

## [Decision Letter · Decision Letter 1]

5 Nov 2024

Curcumin modulated gut microbiota and alleviated renal fibrosis in 5/6 nephrectomy-induced chronic kidney disease rats

PONE-D-24-25478R1

Dear Dr. Tian,

We’re pleased to inform you that your manuscript has been judged scientifically suitable for publication and will be formally accepted for publication once it meets all outstanding technical requirements.

Kind regards,

Md. Jamal Uddin

Academic Editor

PLOS ONE

Additional Editor Comments (optional):

Reviewers' comments:

Reviewer's Responses to Questions

**Comments to the Author**

1. If the authors have adequately addressed your comments raised in a previous round of review and you feel that this manuscript is now acceptable for publication, you may indicate that here to bypass the “Comments to the Author” section, enter your conflict of interest statement in the “Confidential to Editor” section, and submit your "Accept" recommendation.

Reviewer #1: All comments have been addressed

2. Is the manuscript technically sound, and do the data support the conclusions?

Reviewer #1: Yes

3. Has the statistical analysis been performed appropriately and rigorously? 

Reviewer #1: Yes

4. Have the authors made all data underlying the findings in their manuscript fully available?

Reviewer #1: Yes

5. Is the manuscript presented in an intelligible fashion and written in standard English?

Reviewer #1: Yes

6. Review Comments to the Author

Reviewer #1: (No Response)

7. PLOS authors have the option to publish the peer review history of their article (what does this mean?). If published, this will include your full peer review and any attached files.

Reviewer #1: No

---

## [Editor Report · Acceptance letter]

8 Nov 2024

PONE-D-24-25478R1 

PLOS ONE

Dear Dr. Tian, 

I'm pleased to inform you that your manuscript has been deemed suitable for publication in PLOS ONE. Congratulations! Your manuscript is now being handed over to our production team.

Kind regards, 

on behalf of

Dr. Md. Jamal Uddin 

Academic Editor

PLOS ONE